# About the Dominance of Mesopores in Physisorption in Amorphous Materials

**DOI:** 10.3390/molecules26237190

**Published:** 2021-11-27

**Authors:** Christoph Strangfeld, Philipp Wiehle, Sarah Mandy Munsch

**Affiliations:** Bundesanstalt für Materialforschung und-Prüfung (BAM), Unter den Eichen 87, 12205 Berlin, Germany; philipp.wiehle@bam.de

**Keywords:** physisorption, mesopores, amorphous materials, macropores, adsorbed water layer thickness, material moisture, moisture distribution

## Abstract

Amorphous, porous materials represent by far the largest proportion of natural and men-made materials. Their pore networks consists of a wide range of pore sizes, including meso- and macropores. Within such a pore network, material moisture plays a crucial role in almost all transport processes. In the hygroscopic range, the pores are partially saturated and liquid water is only located at the pore fringe due to physisorption. Therefore, material parameters such as porosity or median pore diameter are inadequate to predict material moisture and moisture transport. To quantify the spatial distribution of material moisture, Hillerborg’s adsorption theory is used to predict the water layer thickness for different pore geometries. This is done for all pore sizes, including those in the lower nanometre range. Based on this approach, it is shown that the material moisture is almost completely located in mesopores, although the pore network is highly dominated by macropores. Thus, mesopores are mainly responsible for the moisture storage capacity, while macropores determine the moisture transport capacity, of an amorphous material. Finally, an electrical analogical circuit is used as a model to predict the diffusion coefficient based on the pore-size distribution, including physisorption.

## 1. Introduction

There is a wide variety of natural and man-made porous materials. Amorphous porous materials represent the largest share of porous materials. The pore network of these materials ranges from pore widths of a few nanometres to several hundreds of micrometres [1]. Consider first the man-made materials: in 2020, the global man-made mass exceeded all living dry biomass [2] and is likely to exceed the wet biomass in 2037 [2]. Since 1900, the share of so-called anthropogenic mass has been growing exponentially [3]. Thus, ∼20% of this anthropogenic mass was produced in the last decade alone [2]. Concrete accounts for the largest share at ∼40%, followed by aggregates at ∼35%, bricks at ∼10%, and asphalt at ∼5% [2]. Thus, about 90% of all man-made materials are directly related to human building activities. Manufacturing building materials causes about 11% of global CO${}_{2}$ emissions [4], with the cement industry accounting for the largest share at around 8% of global CO${}_{2}$ emissions [5]. Not only the construction but also the maintenance of these structures requires a high input of resources. For example, around 71% of all repair costs for reinforced concrete are caused by degradation due to corrosion, mainly due to moisture penetration, chloride migration, and carbonation [6]. The cumulative yearly investment in the maintenance of the transport infrastructure in Germany amounts to 13 billion euros [7]. Thus, in Germany, approximately 9 billion euros in repair costs are generated by mass transfer within the amorphous, porous cement matrix. Extrapolated to the EU, the figure is approximately 36 billion euros. This trend is also reflected in the waste industry: 36.4% of all waste of the EU comes directly from the building industry and a further 25.3% from mining and aggregates production [8]. The figures are similar in other highly developed countries.

Besides construction materials, the second, even larger, amorphous medium is the materials of our earth ground. This includes soils such as sands and clays but also rocks such as sandstone, limestone, and granite. Even though soils and rocks differ greatly in terms of stiffness, colour, porosity, density, etc., they all nevertheless contain an open pore system with pore-sizes in the nanometre and micrometre ranges, comparable to those of building materials. This pore network governs processes such as the penetration of nitrates into soils [9,10], the emission of hazardous radon from rock layers [11], and the CO${}_{2}$ release from permafrost and traditional soils [12].

Humans are surrounded by amorphous materials. The prediction of material moisture and moisture transport is of major importance. Ions migrate through the adsorbed water film inside the pore network into the materials, which can lead to harmful deterioration, such as corrosion of reinforced concrete or chemical contamination of soils. Furthermore, moisture itself can be the object of interest. For example, 16% of the global area is facing a high or very high risk of desertification [13]. Being able to predict evaporation and transpiration rates of soils is important for climate models [14,15,16]. Furthermore, by 2050, more than half of the population (52%) will live under stressed water resource conditions [17].

Although material moisture and moisture transport are of great importance in amorphous porous media, knowledge about their modelling is low due to their high complexity [18,19]. In the hygroscopic range, the pore network is only partially saturated. Both hydraulic conductivity and diffusion coexist and contribute to mass transfer [20,21,22]. First, a sorption theory is required for physisorption of hydrophilic materials. One of the first adsorption theories by Langmuir in 1918 [23] only modelled monolayer adsorption and is only valid at very low humidity. In 1938, the popular BET adsorption theory by Brunauer–Emmett–Teller was introduced [24]. Within this theory, monolayer and multilayer adsorption can be modelled. This theory is thus able to cover the hygroscopic range and enables the derivation of entire sorption isotherms. Nevertheless, the BET theory only applies to plane surfaces. However, the pore network of amorphous materials consists not only of slit-shaped pores but also of arbitrary pore geometries, such as cylindrical or spherical pores, so that curvature effects of the water film have to be taken into account, especially inside pores in the low nanometre range. Therefore, Hillerborg implemented the Kelvin equation [25,26] into the BET theory [27] in 1985. Based on this approach, the water layer thickness can be computed for arbitrary pore radii and humidity levels. This approach was used by Ishida et al. to predict the material moisture of cement-based samples stored in a climate chamber [28,29,30]. This sorption theory enables the computation of moisture-related effects such as concrete shrinkage, thermal conductivity, and hysteresis of the sorption isotherm. With the help of humidity sensors embedded directly into cement-based materials, the humidity can be measured and the local material moisture calculated [31]. This has been the basis to solve the entire mass balance for moisture in porous media and to derive the diffusion coefficient and the hydraulic conductivity without the need of a diffusion cell [22].

In the last two decades, several advances have been made regarding theoretical and experimental determination of moisture transport in amorphous materials. Nonetheless, further research is required. Although the pore-size distribution (PSD) can be measured, the exact geometry and connectivity of the pore network remain unknown for most materials. Due to the great variation in pore size, different physical effects of mass transfer coexist: hydraulic conductivity, molecular diffusion, Knudsen diffusion [32,33], inkbottle water [31,34], etc. They all have a significant effect, so that a separate quantification is not possible, and thus the prediction of moisture transport becomes very complex. However, as a first step in modelling moisture transport, the distribution of material moisture in the pore network needs to be known.

Precise models for material moisture distribution and mass transfer in porous media posssess ever-increasing importance. The above-mentioned aspects in terms of CO${}_{2}$ emissions and resource consumption are leading to a rethink in the building industry and raising the demand for sustainable building materials such as earth. However, the mechanical properties of earth-building materials are significantly affected by the relative humidity (RH), which must be taken into account in the design codes to guarantee structural reliability [35]. Strength and stiffness decrease with increasing RH depending on the material composition, which is discussed in detail elsewhere [36,37,38]. Models to predict moisture uptake, storage, and transport in earth-building materials are needed to reliably estimate the load-bearing capacity of such structures, enabling wide application based on a standardised design code. Besides earth-building materials, other cement-reduced or cement-free building materials are attracting more attention [39,40]. A step beyond is the circular construction industry [4], including new sustainable additives [41] as well as recycling of existing construction materials [42]. Reaching an entirely sustainable economy without net-emission of CO${}_{2}$, soils and their fluid dynamic properties become more important. Advanced water management relies heavily on these properties to reduce water stress [17], tree mortality [43], and global desertification vulnerability [13], based on optimal irrigation [44]. Furthermore, reducing soil sealing in urban areas [45] promotes self-cooling cities [46], representing a huge potential for energy saving [47]. Moreover, physisorption is crucial for any combination of adsorbate and adsorbent. Mesoporous molecular sieve materials such as MCM-41 have huge inner surface areas of up to 600 m${}^{2}$/g and show high potential for moisture regulation due to physisorption [48]. The capability of MCM-41 regarding CO${}_{2}$ separation from gas mixtures is also influenced by the material moisture [49]. Similar to MCM-41, hydrophilic metal-organic frameworks (MOF) possess large inner surfaces and their physisorption characteristics significantly influences the efficiency of adsorption heat pumps and atmospheric water generators [50]. MOFs were tested with other adsorbates such as methanol and ethanol with the purpose to optimise the purification of alcohol-based biofuels [51]. Physisorption of moisture also influences the permeability of CO${}_{2}$ and N${}_{2}$ in zeolite membranes. The interactions of material moisture and the efficiency of gas separation are not fully understood [52]. Additionally, mesoporous germanate is used for CO${}_{2}$ separation and, depending on the configuration, possesses a bimodal pore size distribution [53]. Due to distinct changes in electrical impedance due to physisorption of water vapour, mesoporous materials such as metal phosphates might be used directly as humidity sensor [54]. All these rising challenges require precise models for material moisture and mass transfer in porous media. Therefore, we chose a general approach that is adaptable to all mentioned materials. This contributes to an application-driven material design.

In this study, three different amorphous porous materials were investigated. Based on the measured PSD, Hillerborg adsorption theory was used to compute the moisture distribution for different pore geometries, such as slit-shaped, cylindrical, and spherical pores. These geometries represent idealised models to describe real and very complex pore networks. Thus, it was necessary to validate which pore geometry best describes the real pore network. This was done by means of experimentally measured sorption isotherms. Knowing the PSD and the pore geometry, the water layer thickness could be calculated for all pores in the hygroscopic range. Mesopores were found to contain the most material moisture, although the pore network was strongly dominated by macropores. Based on this result, an extension of the electrical analogical circuits to model the diffusion coefficient of porous materials was discussed.

## 2. Theory

Amorphous materials have a broad PSD from the low nanometre range to hundreds of micrometres. To describe the underlying physics of material moisture and mass transfer in these materials, a classification of the pore-sizes is reasonable. Furthermore, the pore geometries investigated are discussed below. Subsequently, the prediction of the raising water layer thickness due to physisorption is explained. For the entire hygroscopic humidity range, the influence of pore geometry is shown for different pore widths.

### 2.1. Pore Classification and Geometry

The pore network of amorphous materials consists of very different pore-sizes, ranging from approximately 2 nanometres in diameter up to hundreds of micrometres. Depending on the pore-size, different physical effects become prominent. Therefore, a grouping of the pores is useful. In this study, we follow the IUPAC manual, which was introduced in 1985 [55] and updated in 2015 with two additional physisorption isotherms [56]. The classification is as follows:-Macropores: Pore width exceeds 50 nm;-Mesopores: Pore width is between 2 nm and 50 nm;-Micropores: Pore width does not exceed 2 nm.

In this study, three different pore geometries are analysed: slit-shaped pores, cylindrical pores, and spherical pores. A slit-shaped pore is defined as two infinite, parallel plates that form the corresponding pore volume. The so-called pore width is the distance between the two parallel pore fringes. For cylindrical pores, the pore width is the pore diameter. This pore is assumed to be infinitely long to avoid any three-dimensional boundary effects. For spherical pores, the pore width is the pore diameter. In the following, the term pore radius is used for all three geometries for comparison. For the slit-shaped pore, the pore radius is half the distance of the two parallel pore fringes. For cylindrical and spherical pores, the radius is half the pore diameter.

In amorphous materials, several pore-sizes coexist in the material. In such pore networks, hysteresis occurs between adsorption and desorption. Several effects causing this phenomenon are discussed in [57]. One main reason is the pore blocking during desorption. Larger pores cannot release their pore water because they are connected only via smaller pores to the network, which remain saturated. Depending on the PSD, pore blocking, or the so-called inkbottle water, mesopores may account for more than 64% of the total material moisture [31]. However, only adsorption will be discussed in the following.

### 2.2. Hygroscopic Water Adsorption

To predict physisorption or sorption isotherms of different PSDs, a sorption theory is required. This theory must be able to capture multilayer adsorption, various pore-sizes, and different pore geometries. The often-used BET theory [24] is an extension of the Langmuir adsorption theory, which only allows for the formation of a monolayer of adsorbed molecules [23]. Although BET theory allows for random multilayer adsorption, it is only valid for plane surfaces [24] and only slit-shaped pores can be analysed. In order to compute adsorption also on curved surfaces, Hillerborg [27] incorporated the Kelvin equation [25,26] into the BET theory. This allows the analysis of arbitrary convex or concave pore geometries, thus also of cylindrical and spherical pores [27]. The Kelvin equation is considered to be applicable down to a capillary diameter of 1 nm [58]. Originally, Hillerborg only considered single pores with a distinct width. However, several authors applied this approach to full PSDs of amorphous media, including non-plane pore geometries [22,28,30,31,33,59].

Hillerborg’s theory and its adaption to PSDs is briefly summarised here. Equation (Equation 1) gives the computation of the adsorbed water layer thickness $t_{a}$ in m [27]. Thereby, *C* is a material constant related to the heat of adsorption in the first layer [24]. $t_{w}$ in m is the thickness of one monomolecular water layer. *R* in J mol${}^{-1}$ K${}^{-1}$ is the ideal gas constant, *T* in K is the temperature, *M* in kg mol${}^{-1}$ is the molecular mass of water, γ in Nm${}^{-1}$ is the surface tension of liquid water, and $r_{1}$ and $r_{2}$ in m are the pore radii. The humidity at which the air volume in the centre of a pore disappears is the so-called maximum humidity $h_{m}$, defined in Equation (2) [27]. In the case of plane surfaces, $h_{m}$ remains at a value of one, which resembles the BET theory. However, in a partially saturated state, $t_{a}$ and $h_{m}$ both depend on the chosen pore geometry.
(1)$$\begin{array}{ccc} t_{a} & = & \frac{t_{w}hC}{\left(1-\frac{h}{h_{m}}\right)\left(1-\frac{h}{h_{m}}+Ch\right)} \end{array}$$
(2)$$\begin{array}{ccc} h_{m} & = & \exp\left(-\frac{\gamma M}{\rho_{l}RT}\left(\frac{1}{r_{1}-t_{a}}+\frac{1}{r_{2}-t_{a}}\right)\right) \end{array}$$

Finally, Hillerborg’s approach incorporates water film moisture and capillary moisture and is therefore able to predict the material moisture for the entire sorption isotherm at varying pore-sizes and geometries [27]. The physical quantities and the input parameter are summarised in Table 1.

### 2.3. Water Layer Thickness of Partially Saturated Pores

Based on Equations (Equation 1) and (2), a water layer thickness can be calculated for every relative humidity (RH) and every pore-size. Furthermore, the curvature and hence the pore geometry also influence the amount of adsorbed water. The smaller the two pore radii $r_{1}$ and $r_{2}$ are, the higher the adsorbed water layer thickness [27,31]. This is illustrated in Figure 1 for three different humidity levels of 40% RH, 75% RH, and 90% RH, and for the three pore geometries: slit-shaped pore, spherical pore, and cylindrical pore.

A higher humidity level leads to a higher water layer thickness for all geometries. For the slit-shaped pore, the layers have a thickness of around 0.5 nm, 1.4 nm, and 3.5 nm, respectively. Pores smaller than twice the water layer thickness are already completely saturated. Thus, for example, at 90% RH, the water layer thickness increases until a pore width of 7 nm is reached (the water layer simultaneously starts at both sides of the pore). Above 7 nm, a partial saturation is present. The layer thickness does not increase any more, as it is independent of the pore geometry. In contract, the water layer of cylindrical and spherical pores is dependent on the geometry. In the case of partial saturation, the initial pore radius in Equation (2) is further decreased due to the rising water layer $t_{a}$. This increased curvature of the air–water interface leads to a reduction in the water vapour partial pressure, allowing small pores to form thicker water layers compared to slit-shaped pores. For example, at 75% RH, the maximum layer thickness without water layer curvature is 1.4 nm. For cylindrical pores, it is 6.1 nm and for spherical pores 8.7 nm. This is the upper limit for full saturation. In larger pores, partial saturation occurs. The curvature of the water layer still has an effect but decreases quickly. For pores larger than 1 μm in diameter, slit-shaped, cylindrical, and spherical pores have nearly the same water layer thickness and curvature effects can be neglected.

As shown in Figure 1, pores with a diameter between 4 nm to 48 nm are mainly responsible for the increased pore saturation in the broad humidity range of 40% RH ≤h≤ 90% RH, compared to slit-shaped pores. In summary, mesopores are influenced the most by pore geometry.

## 3. Materials and Methods

Three material types were investigated as representatives of the amorphous materials. These were sandstone, earth building material, and screed, which are described in detail below. Subsequently, the analytical methods to measure the PSDs are discussed. To validate the chosen pore geometry, sorption isotherms were requested. The corresponding experiments are therefore briefly summarised.

### 3.1. Sandstone

Kylltal (Kyllburg) sandstone is mined in the Eifel region in Germany and is part of the upper Buntsandstein formation. It is fine- to medium-grained with a clayey, partially clayey-ferritic cement [60], and its colour varies from red-yellow to brown. Special features represent brown-yellow parallel stratifications as well as clay lenses that occur locally [61]. The homogeneous and weathering-resistant sandstone consists mainly of quartz (48%) and stone fragments (37%) [60,62]. The porosity was determined to be 16.63% by mercury intrusion porosimetry (MIP). In comparison, in Grimm [60] the porosity is stated to be 13.35%.

The Brenna sandstone is found in Brenna, Cieszyn, and Katowice in Poland. It is grey in colour, consists mainly of quartz and feldspar, and has only a small amount of clayey matrix. Other characteristic elements are uniformly distributed glauconitic aggregates [63,64]. As the fine-to-medium sized grains have numerous contact points, it seems to be well compacted. Consequently, its porosity is low, with a value of 6.29% obtained by MIP measurements.

The sandstone samples in this study had a cylindrical shape, with a diameter of 20 mm and a height of 70 mm to 100 mm. As the sandstone samples were saturated in desiccators with different relative humidities, the authors prepared two sister samples for each desiccator/saturation stage.

### 3.2. Screed

In a previous study, eight screed types and their moisture transport were investigated based on embedded humidity sensors [31]. From this batch, two screeds were selected for the current study: a cement-based screed and a calcium-sulphate-based (CS) screed.

The cement-based screed is a concrete screed with a compressive strength of 35 Nm m${}^{-2}$ and a flexural strength of 5 Nm m${}^{-2}$, according to [65]. The recommended water demand is 0.11 L kg${}^{-1}$, and the resulting consistency class is F1 to F2, following [66]. The aggregate size is 0 mm to 8 mm. This concrete screed is designed for indoor and outdoor application. It is waterproof and frost-resistant.

The calcium sulphate screed is a floating screed with a compressive strength of 25 Nm m${}^{-2}$ and a bending tensile strength of 5 Nm m${}^{-2}$, according to [65]. The recommended water demand is 0.163 L kg${}^{-1}$, and the resulting consistency class is F2 to F3, following [66]. The aggregate size is 0 mm to 4 mm. This floating screed is self-levelling and only applicable indoors. It has a high heat conductivity, which makes this screed suitable for underfloor heating.

The weights of the dry screeds and the water were checked by a high precision balance before mixing. After concreting, the samples were covered and stored over-night in the production room (at approx. 295 K). On the following morning, the samples were stored in a ventilated climate chamber at an ambient relative humidity of h=50% RH and an ambient temperature of T=296 K. Further details are discussed in [31,67].

### 3.3. Earth-Building Materials

The materials were sourced from local manufacturers of prefabricated earth-building products. The earth mortar was provided as premixed dry mortar with a compressive strength of 2 Nm m${}^{-2}$ according to the German standard [68]. The mortar was adjusted to a consistency class or spread diameter of 175 mm and showed a shrinkage of 2% and a water demand of 0.15 L kg${}^{-1}$. The bulk density of the mortar was determined to be 1.95 kg L${}^{-1}$. The aggregate size ranged from 0 mm to 4 mm, with the sand fraction (0.063 mm to 2.0 mm) clearly dominating at 66 wt.%, followed by silt at 26.28 wt.% (0.002 mm to 0.063 mm) and clay as a binder at 6.67 wt.% (<0.002 mm). The gravel fraction was solely 1.05 wt.%. This grain-size distribution was determined according to [69] and represents a typical distribution for earth mortar, leading to good processability. The semi-quantitative X-ray diffraction analysis showed medium amounts of illite and muscovite and minor amounts of chlorite and vermiculite. The total organic matter content was determined by ignition loss according to [70] and amounted to 1.86 wt.%.

The earth blocks are perforated extruded blocks in 3DF format according to the German masonry format (240 mm × 175 mm × 113 mm). The mean compressive strength was determined to be 5.38 Nm m${}^{-2}$, so that the blocks could be classified in compressive strength class 4 according to [71]. The bulk density of the blocks was 1.87 kg L${}^{-1}$, and the aggregate size 0 mm to 4 mm. Unlike the mortar, the grain-size distribution was more homogeneous and contained notably higher fractions of clay (16.05 wt.%) and silt (43.90 wt.%) and a lower fraction of sand (31.82 wt.%). The gravel fraction was higher than the gravel fraction of the mortar at 5.23 wt.%. The semi-quantitative X-ray diffraction analysis showed a high amount of illite, medium amounts of chlorite and muscovite, and a minor amount of vermiculite. The total organic matter content was 6.29 wt.%, which is significantly higher than the mortar. This is due to the addition of cellulose-based fibres leading to higher ductility and dimensional stability in the prefabricated earth blocks.

### 3.4. Mercury Intrusion Porosimetry and Gas Adsorption

Three types of amorphous porous material were investigated and their PSDs quantified. Two methods were used to capture macro- and mesopores, MIP, and gas adsorption [72,73]. In MIP, pressures between 0.01 MPa and 400 MPa were generated during the tests by the measurement device ‘MicroActive AutoPore V 9600’. The conversion of pressure to a certain pore radius was done with the Washburn equation [74]. The measurement procedure was the exact reproduction of the international standard [72]. The calculated pore radius started at approximately 50 μm, and the sensitivity was recorded down to a minimum pore radius of approximately 2 nm. The dry sample masses used were between 0.8 g and 1.5 g, and the model assumed cylindrical pores to convert the pressure into pore-sizes. The other method used was the gas sorption [73,75] based on physisorption of nitrogen gas at 77 K in a pressure range of 4.5 mbar to 1 bar. The measurements were performed by the device ‘ASAP 2010 V5.03’. The conversion from pressure to a certain pore diameter followed the Barrett–Joyner–Halenda (BJH) theory by assuming cylindrical pores [76]. The BJH theory includes the layer thickness of the adsorbed nitrogen according to Halsey [77] and the Kelvin equation for calculating the pore radius. The sample mass varied between 1 g and 4.5 g. The measurement procedure was the exact reproduction of the international standard [75]. The measured pore radii were between 0.8 nm and 100 nm. For small pore radii of 0.9 nm to approximately 10 nm, gas adsorption showed significantly higher sensitivity than the MIP.

### 3.5. Dynamic Vapour Sorption and Desiccator

Samples of earth-building materials were measured using a Gravisorp 120 multisample dynamic vapour sorption (DVS) to provide detailed sorption isotherms, especially in the range between a relative humidity of *h* = 30–80% RH. Samples with a weight of 15 g to 20 g were randomly taken from stones and mortars. One sister sample was prepared for each material. The samples were preconditioned at h=50% RH and 296 K and subsequently desiccated in the DVS at 293 K. Thereupon, the measurement was carried out from h=0% RH to h=95% RH in steps of 5% RH at a constant temperature of 293 K. A total of three adsorption-desorption cycles were performed.

The sandstone samples were partially saturated by adsorption in desiccators that had specific relative humidity levels. Six different relative humidities were regulated using salt solutions that are listed in Table 2. The relative humidity in the desiccators was controlled by low-voltage humidity sensors placed inside the desiccators via the openings of the lids. The samples were stored in the desiccators until they achieved mass constancy according to [78]. Therefore, the sample weights were checked in regular time intervals using a digital balance with a verification scale of 0.1 g and a readability of 0.01 g.

## 4. Results

First, the PSD of the six samples and the amount of mesopores are discussed. This is the required input for Hillerborg’s adsorption theory to compute the water layer thickness and the material moisture. The chosen pore geometry is validated by means of experimentally measured sorption isotherms. It was shown that most of the material moisture is located in mesopores. Eventually, the consequences of these new findings for modelling diffusion coefficients including physisorption are discussed.

### 4.1. Measured Pore-Size Distribution

Three types of different materials were investigated experimentally, including two different types of each material. Since the PSD mainly influences the material moisture and the mass transfer inside the pore network, it needs to be known. The cumulated PSD as a combination of MIP and gas adsorption is shown in Figure 2. Gas adsorption is more sensitive to the full mesopore range, while MIP is able to measure larger mesopores and macropores. A threshold of 16 nm in diameter was chosen for the combination of these two distributions. Below this threshold, gas adsorption data were used, above it, MIP data. Furthermore, the gas adsorption data were refined in the post-processing. The measured pore volume at one sampling point is equally distributed to ten sub-sampling points between the upper and lower diameter. Without this refinement, the sorption isotherm appears more like a step function due to the coarse resolution.

As shown in Figure 2, all six material samples were amorphous materials with pore diameter ranging from 1.8 nm to more than 10 μm. The total pore volume varied between 26.3 mm${}^{3}$ g${}^{-1}$ and 166.4 mm${}^{3}$ g${}^{-1}$. Besides the total pore volume, the PSDs also showed different trends, especially for the two earth materials. Although the earth block had the highest total pore volume of all six samples, the earth mortar showed the largest number of pores in the range between 300 nm and 10 μm in diameter. Considering the screed samples, the calcium sulphate-based screed has almost twice the total pore volume of the cement-based screed, but the calcium sulphate-based screed had almost no pores smaller than 12 nm in diameter. Table 3 depicts the amount of mesopores in more detail and determines the absolute and relative amount of mesopores. As already indicated by the PSD, the cement-based screed with 26.9% mesopores had more than five times as many mesopores as the calcium sulphate-based screed with 5.2%. The earth mortar with a high amount of macropores possessed only 7.7% mesopores. Considering the sandstones, those from Brenna and Kylltal had a similar absolute amount of mesopores, although they differed considerably in both the total pore volume and the relative amount of mesopores. These six PSDs were used to predict the water layer thickness and the cumulated saturation to yield the theoretical sorption isotherms for the different pore geometries.

The amount of mesopores differs significantly, e.g., by 17.2% between the calcium sulphate-based screed and the Brenna sandstone. Consequently, the question arises what causes these great differences. There is not a unique influencing factor. Instead, the authors speculate that the complex combination of chemical composition and forming condition determines the pore network (sedimentation and diagenesis conditions of the sandstones, hydration process of the screed and earth material, etc.). In the case of screed, we think that the formation of calcium silicate hydrates may lead to a totally different pore geometry and network compared to sandstones. However, detailed analysis of the chemical composition is out of the scope of this study.

### 4.2. The Sorption Isotherm and Its Dependence on the Pore Geometry

The PSD was measured for the six samples. Thus, the pore saturation can be computed for every pore diameter and every humidity level using Equations (Equation 1) and (2). The cumulation of saturation of all pores yields the material moisture, i.e., the sorption isotherm [22,31]. Thereby, different pore geometries showed different saturation levels, which led to varying sorption isotherms. The higher the pore curvature, the higher the saturation. Therefore, spherical pores always led to the highest material moisture, slit-shaped pores to the lowest, and cylindrical pores always lay between these two. This is shown for the Kylltal sandstone and the earth block in Figure 3. The total porosity of the earth block was more than twice that of the Kylltal sandstone. Therefore, the predicted material moisture is higher at every relative humidity. Furthermore, the measured sorption isotherm was included as well. The measured values of the Kylltal sandstone highly correlated to the predicted sorption isotherm of slit-shaped pores. In contrast, the earth block matched spherical pores best. Of course, this is an idealisation. Not all the pores will have a perfect sphere or slit. The real pore system will be formed more or less chaotically, maybe with a tendency towards a certain pore configuration. On the other hand, without the assumption of a pore geometry, the water layer thicknesses and material moisture cannot be calculated. Therefore, an assumption regarding the pore geometry is required based on the best fit between the measured sorption isotherm and the three predicted ones.

Table 4 summarises which pore geometry yields the highest correlation to the measured values for the six material samples. Both materials with clay minerals, earth block, and mortar correlate to spherical pores. The two screeds correlate to cylindrical pores, which is consistent with previous studies that analysed eight screed types [31]. In the case of sandstone, the pore geometry is more diverse with slit and cylindrical pores. Other sandstones such as Schönbrunn and Bozanov sandstone most closely match the spherical pores [79]. However, as the pore geometry is unknown a priori, validation is required. This was done experimentally based on the sorption isotherm. With the knowledge of the pore geometry, the water layer thickness and the material moisture can be calculated appropriately.

The fitting of the pore geometry is just an ordinary correlation without the incorporation of the material physics. This is mainly caused by the fact that the “true” pore system is unknown. However, within the group of the sandstones, the bet fitting pore geometry varies. One speculative explanation is that the Kylltal sandstone contains mica minerals and a large amount of clayey binders. In scanning electron microscope results, these minerals look more rodlike and therefor might cause a more slit-shaped pore geometry [60]. However, a detailed discussion between assumed pore geometry and tomographic imaging is outside the scope of this study.

### 4.3. Moisture Distribution versus Pore-Size Distribution in Dependence on the Pore Geometry

The PSD was measured and the pore geometry determined via the sorption isotherm. Thus, for a certain humidity level, the water layer thickness can be computed for every pore-size. The water layer thickness multiplied by the pore volume yields the total amount of water held by the considered pore-size. On the one hand, the cumulation of all pores yields the sorption isotherm; on the other hand, this analysis provides the relative moisture distribution as a function of the pore-sizes. This allows a calculation of the amount of water, held by mesopores in the sample.

Figure 4 shows the cumulated amount of water versus the pore diameter. If the water held in all pores is cumulated, the total amount of water in the sample is obtained. This corresponds to the material moisture at this humidity level, which is already expressed by the sorption isotherm. Thus, for a certain humidity level, the material moisture differs between the investigated pore geometries. However, starting the cumulation at the smallest diameter of 1.8 nm, one can see how much water is held in the pores. Thereby, spherical pores would show higher amounts of water due to the thicker water layer caused by the increased curvature. For a better comparison between samples, the cumulated amount of water was normalised by the total amount of water for each pore geometry of each sample at the considered humidity level. This is shown in Figure 4 for a relative humidity of 40% RH. The calcium sulphate-based screed had the lowest amount of mesopores and the cement-based screed the highest, see Table 3. The Kylltal sandstone had an medium amount of mesopores but is the only sample with slit-shaped pores according to Table 4. Nevertheless, the computations for comparison were done for all three pore geometries. As can be seen from Figure 4, the pore geometry has only a minor influence on the water distribution at this humidity level. In the case of the Kylltal sandstone, 50% of the material moisture was held by pores with a diameter below 6 nm. For the two screeds, half of the moisture was reached at diameters of around 12 nm and 28 nm, respectively.

The same computation was performed for a humidity level of 90% RH, as shown in Figure 5. Here, the influence of the pore geometry becomes prominent. In the case of slit-shaped pores, the material moisture was redistributed towards smaller pores. At first, this result may seen contradictory, but as shown, the higher the curvature, the higher the pore saturation. Therefore, spherical pores show a higher saturation as slit-shaped pores, as illustrated in Figure 1. However, the higher saturation also increased the total amount of water in the sample. This is expressed by the sorption isotherms for the different pore geometries in Figure 3. In absolute numbers, spherical pores always adsorb more water than slit-shaped pores. In contrast, Figure 5 shows the relative distribution of the material moisture. For example, slit-shaped pores in Kylltal sandstone at 90% RH generated a material moisture of 0.55 wt.% and spherical pores 1.22 wt.%. Although spherical pores hold more water in absolute numbers, their relative proportion decreases due to the chosen normalisation.

Considering the cylindrical and spherical pores in Figure 5, it can be seen that they are highly correlated. The difference between these two geometries was below 10 nm in most cases. Furthermore, the differences between the three geometries are negligible for pores larger than 1 μm in diameter. Although at 90% RH the cumulated amount of water shifted to larger pores compared to 40% RH, the main statement remains: most of the moisture is located in mesopores. Even the calcium sulphate-based screed with the lowest portion of mesopores (5.2%) held more than 64% of the moisture in pores smaller than 50 nm in diameter.

### 4.4. Moisture Distribution

At low humidity of around 40% RH, the influence of pore geometry is almost negligible but becomes significant at high humidity of around 90% RH. However, due to the experimental validation based on the sorption isotherm, the best-fitting pore geometry was determined, see Table 4. For each material with its corresponding pore geometry, the moisture distribution versus pore diameter is shown in Figure 6, Figure 7 and Figure 8 for 40% RH, 75% RH, and 90% RH, respectively.

At 40% RH, the sandstone and the earth-building material samples were close to each other. More than 88% of the moisture was located in mesopores in all four samples. The two screed samples deviated from the other four samples, especially the calcium sulphate-based screed. As shown in Table 3, only 5.2% of the entire pore network are mesopores. Obviously, the lower the amount of mesopores, the less moisture can be located in these pores. Nevertheless, more than 67% of the material moisture was still held by mesopores.

At a corresponding relative humidity of 75% RH, all lines moved slightly towards larger pores. At this humidity, pores with widths below 3 nm were already fully saturated, independent of the pore geometry [27,31]. The Kylltal sandstone held most of the moisture in small pores, because it was the only sample with slit-shaped pores. Furthermore, the calcium sulphate screed held more than 65% of the moisture in mesopores.

Figure 8 shows the moisture distribution for 90% RH. This was close to the upper limit of the hygroscopic range at around 96% RH. However, the overall trend between 75% RH and 90% RH was quite similar. For a better comparison, the results are resumed in Table 5. The gas adsorption used to measure the PSD also detected pore-sizes below 2 nm. In most cases, the pore-sizes started at 1.7 nm. Although these pores between 1.7 nm and 2 nm would already be micropores according to the IUPAC definition used, the moisture contained in these pores is aggregated to mesopores. A differentiation between micro- and mesopores is therefore not performed in Table 5. However, in most cases, at least three quarters of the material moisture was located in mesopores. Only the calcium sulphate-based screed with only 5.2% mesopores in its pore network had lower values. Nevertheless, two thirds of the moisture was located in mesopores or in only 5.2% of the pore network, respectively. Furthermore, the earth mortar reached the highest ratio at 40% RH. In this case, 92.9% of the material moisture was located in mesopores, while these pores contribute only 7.7% to the entire pore network.

## 5. Discussion

Three different materials were investigated, each with two different types of materials. All samples contained pores between approximately 2 nm and 100 μm. The total pore volume varied between 26.3 mm${}^{3}$ g${}^{-1}$ and 166.4 mm${}^{3}$ g${}^{-1}$, whereas the relative proportion of mesopores ranged between 5.2% and 26.9%. Three pore geometries are compared for each sample. The water layer thickness was determined by the sorption theory of Hillerborg [27]. This theory is able to capture flat, convex, and concave pore shapes. The experimental sorption isotherms were used to validate the assignment of the corresponding geometry to each sample. Finally, the moisture distribution over the pore diameters was computed.

Although the share of mesopores is significantly smaller compared to macropores, the mesopores contain most of the material moisture. This general statement holds for all pore geometries and all humidity levels in the hygroscopic range. As shown in Table 5, although mespores makes up only 5.2% of the pore network, they hold around two thirds of the material moisture. The physical explanation for this is the ratio between pore volume and pore surface depending on the pore width. For the following example, two cylindrical pores with the same pore volume are considered. One is a mesopore with a diameter of d=10 nm, and the other is a macropore with a diameter of d=1 μm. It follows that the inner surface of the mesopore is a hundred times larger than that of the macropore. Physisorption always starts at the pore surface. The water layer thickness in the d=1 μm macropore is below 5 nm, even at 90% RH [27]. Furthermore, the water layer thickness increases due to the curvature of the inner surface. This curvature effect is much stronger in mesopores compared to macropores. In our example, the inner surface of the mesopore is one hundred times larger and forms a thicker water layer. For the same pore volume, the water storage capacity of the mesopore with d=10 nm is even more than a hundred times higher than the macropore with d=1 μm. Depending on the humidity, the ratio can even exceed thousand. Mesopores thus determine the material moisture to a large extent, although their pore volume is small compared to the overall pore volume.

Mesopores hold most of the material moisture within the hygroscopic humidity range. Although this gives a deep insight into the moisture distribution in amorphous materials, it does not contain direct information about diffusive moisture transport within the pore network. For the pore widths discussed here, molecular diffusion and Knudsen diffusion coexist, and both have to be taken into account [18,32,80,81,82,83]. Molecular diffusion deals with the molecule-molecule interaction in gases and can be described mathematically by Fick’s law [84]. On the contrary, Knudsen diffusion handles the molecule-pore wall interaction and is described by the Knudsen effect [80,85,86]. This effect occurs if the pore width is comparable with the free path length of the gases in the open space of the pore network. For amorphous materials, is is assumed that the transient state begins approximately when pore radii are below 30 nm to 200 nm [81,87,88,89,90]. In pores with a width of 1 μm, the Knudsen effect leads to a reduction of the molecular diffusion of around 3%. Below pore widths of 60 nm, the reduction of the molecular diffusion is more than 50% [33]. The collision of gas molecules with the pore wall reduces the effective gas diffusion. Consider again the cylindrical macropore with d=1 μm and the mesopore with d=10 nm. In dry pores, the Knudsen effect reduces the effective gas diffusion in the macropore with d=1 μm by around 6.2%. On the other hand, the Knudsen effect reduces the effective gas diffusion in the mesopore with d=10 nm by around 87% [33]. The diffusive moisture transport through the macropore is therefore more than seven times greater than in the mesopore. In non-dry pores, the water layer thickness further decreases the open space for gas diffusion due to physisorption. Taking this into account, the moisture transport capacity of the macropore can be more than ten times higher than in the mesopore, depending on the humidity [33]. This leads to the fundamental insight that in amorphous materials, on the one hand, mesopores govern the material moisture, i.e., the moisture storage capacity. On the other hand, the contribution of mesopores to the diffusive moisture transport is minor or even insignificant.

In the hygroscopic range, the moisture transport is governed by diffusion and advection, quantified by the diffusion coefficient and the hydraulic conductivity, respectively [20,91]. The latter becomes dominant at higher humidity levels [22], whereas at low humidity levels, mass transport is purely diffusive. In amorphous materials, the pore sizes differ by several orders of magnitudes, so that Knudsen diffusion, molecular diffusion, and viscous flow coexist. Thereby, the diffusion coefficient of a material can be interpreted as the resistance of the material against “normal diffusion”, which would be molecular diffusion in free air [92]. In the context of the dusty gas model, the total or effective resistance of a material against diffusion is the simple summation of the resistance associated with gas-gas collisions and gas-wall collisions [93]. Furthermore, it is assumed that the individual resistances are the reciprocals of the corresponding diffusion coefficients [92,94,95], namely, the bulk diffusion and the Knudsen diffusion. This analogy to electrical resistance enables the design of an equivalent circuit diagram for diffusion. To incorporate hydraulic conductivity as well, the resistance against viscous flow is added to the circuit network, also known as the Schofield model [96]. Different electrical analogical circuits of the dusty gas model and the Schofield model are further discussed in [97,98].

For a single pore, the simple addition of resistance due to molecular diffusion and Knudsen diffusion is an assumption that is debatable. However, these models are too simplistic for an entire pore network. Therefore, Phattaranawik et al. suggest an electrical analogical circuit for pore networks [99]. Beside molecular and Knudsen diffusion, this also considers the transient region. An adaption of these models to building materials seems reasonable for steady-state analysis [93]. Nevertheless, all these circuit networks have been developed mainly to describe membranes in distillation processes. In general, membranes are considered to be thin. Thus, physisorption, although present, is neglected. In contrast, building materials are in general thick structures and physisorption plays an important role, especially for moisture transport. Thus, whichever electrical analogical circuit is chosen, the model for thick structures needs to be extended to include physisorption.

Figure 9a shows a schematic sketch of moisture transport through a thick structure. The side walls are isolated, resulting in one-dimensional moisture transport. On both sides, a potential acts on the porous material. In our case, it is the relative humidity, expressed as the water vapour partial pressure *P*. The potential at the input side is always higher than the potential at the output side $P_{out}<P_{in}$. This gradient generates a moisture flux *J* from the higher potential towards the lower potential. In steady-state, this setup might be sufficiently described by an electrical analogical circuit composed only of resistors. The resistor network only describes the evolution of the diffusion coefficient, which is merely an input parameter to solve an underlying diffusion equation for mass transport, such as Fick’s second law. The diffusion coefficient is thus non-linear linked to the moisture flux via the diffusion equation. The input potential $P_{in}$ increases, which is shown as a stepwise function in Figure 9b. Consequently, the moisture flux $J_{in}$ also increases immediately. However, this is a thick structure including physisorption. Therefore, the incoming moisture flux is first adsorbed at the inner surface of the pore network. Thus, the exiting moisture flux remains constant, although the input potential and the incoming moisture flux are increased. To fulfil the mass balance in such a case, one has to add a sink term *Q* [28,31]. The adsorption process inside the sample can be interpreted as a moisture sink. After some time, more and more moisture is adsorbed until the increased input potential is recognised on the other side. Then, the exiting moisture flux $J_{out}$ also increases. $J_{out}$ continues to increase until the new steady-state is reached. If the output potential is decreased again step by step, the exiting moisture flux $J_{out}$ remains constant first. The adsorbed moisture inside the sample provides the required moisture flux during this desorption process. In this phase, the sample can be interpreted as a moisture source until the new steady-state is reached again.

As shown in Figure 9b, there is a time delay between the incoming moisture flux $J_{in}$ and the exiting flux $J_{out}$. These are dead times, symbolised by $t_{d_{1}}$ and $t_{d_{2}}$. The amplitude of $t_{d_{1}}$ and $t_{d_{2}}$ does not have to be equal and depends on the sample thickness *d* and its physisorption properties. However, an electrical analogical circuit consisting only of resistors could not simulate such a behaviour. Although the new steady-state moisture flux would also initially follow an asymptotic trend due to the underlaying diffusion equation, there would be no dead time. In the case of thin membranes, $J_{out}$ would react as soon as the input potential is modified. However, for thick structures, there is such a delay between input and output moisture flux. To overcome this lack of simulation, an inductor is added after the resistor network, as presented in Figure 10. The resistor circuit itself is adapted from Phattaranawik et al., which was designed for multi-pore-size models [99]. Now, if $P_{in}$ were increased stepwise, the output current would not rise immediately. First, the inductor needs to be loaded, which leads to a delayed reaction of the diffusion coefficient. This in turn is the input for the diffusion equation. Thus, the dead time does not result directly from the inductor but from the delayed reaction of the diffusion coefficient within the diffusion equation. Finally, a dead time also occurs for the moisture flux. This is identical to the behaviour of physisorption of thick structures. The potential, the water vapour partial pressure, increases rapidly. Nevertheless, the material sample needs to be “loaded” first in the form of water adsorption in the pore network. After a certain dead time, the output flux will rise. This analogy also holds for a stepwise potential decrease. The inductor would discharge and the current would remain constant during the dead time. In physisorption, the material would “discharge” by water desorption. Based on this concept of hydraulic inductance, both steady-states and dynamic changes can be simulated and analysed.

This new concept might appear to be a reasonable extension of the existing theory. Nevertheless, the question remains about how to best determine the hydraulic inductance $L_{h}$ of the physisorption process. As shown above for the six material samples, mesopores represent only 5.2% to 27% of the pore network volume but contain between 65% to 93% of the material moisture. However, due to the Knudsen effect, mesopores contribute only slightly to moisture transport [33]. Thus, in a first step, the PSD could be simulated as a kind of binary system. On the one hand, macropores are responsible for moisture transport, but their moisture storage capacity is neglected. On the other hand, the mesopores determine moisture storage capacity and thus the hydraulic inductance $L_{h}$, but do not contribute to moisture transport.
(3)$$\begin{array}{ccc} L_{h} & \propto & f(\frac{V_{meso}}{V_{macro}},\;d) \end{array}$$

The fraction of the volume of mesopores $V_{meso}$ in the volume of macropores $V_{macro}$ could be one of the input parameters for the hydraulic inductance, as shown in Equation (Equation 3). If a high volume fraction of the pore network consists of mesopores, the material would have a high moisture storage capacity. Thus, physisorption would play an important role and $L_{h}$ would be high. Reaching the new equilibrium would be significantly delayed. In contrast, if the material consisted almost entirely of macropores, the moisture transport capacity would be high, while the moisture storage capacity would be low. The resulting hydraulic inductance or physisorption is thus low, and the new equilibrium is reached quickly.

The discussed quantification of hydraulic inductance is based on the assumption of a binary pore system. Of course, this assumption is debatable. However, the derived concept of hydraulic inductance should be considered as a starting point for further theoretical and experimental work regarding mass transfer in porous amorphous materials.

## 6. Conclusions

Amorphous porous materials have a very complex pore network geometry. Thus, several transport mechanisms coexist and influence each other, depending on the pore-size. However, in a first step, the distribution of the material moisture must be known to derive transport coefficients or predict transport processes. Based on the sorption theory of Hillerborg, the water layer thickness of six materials was computed for three different pore geometries. Based on experimental sorption isotherms, the most appropriate pore geometry was validated for each material sample. Using the PSD and the known water layer thickness, the amount of moisture contained in each pore was quantified.

In all samples, macropores dominated the pore network and mesopores only represented between 5.2% to 27% of the available pore volume. Nevertheless, the mesopores hold between 65% and 93% of the material moisture. On the one hand, mesopores contribute only slightly to moisture transport in amorphous materials, mainly due to the Knudsen effect. On the other hand, mesopores govern the moisture storage capacity of a material. Based on these new findings, a model was introduced to predict the diffusion coefficient, including physisorption. An inductor was added to the original electrical analogical circuit by Phattaranawik [99] to account for the physisorption of thick structures. Finally, a first approach to determine the hydraulic inductance purely based on the PSD is discussed.

## Figures and Tables

**Figure 1 molecules-26-07190-f001:**
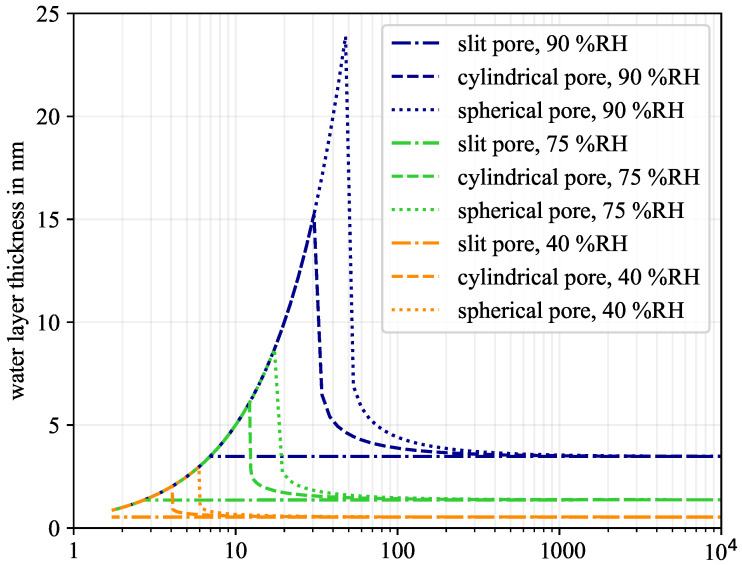
Predicted water layer thickness of different pore geometries due to physisorption.

**Figure 2 molecules-26-07190-f002:**
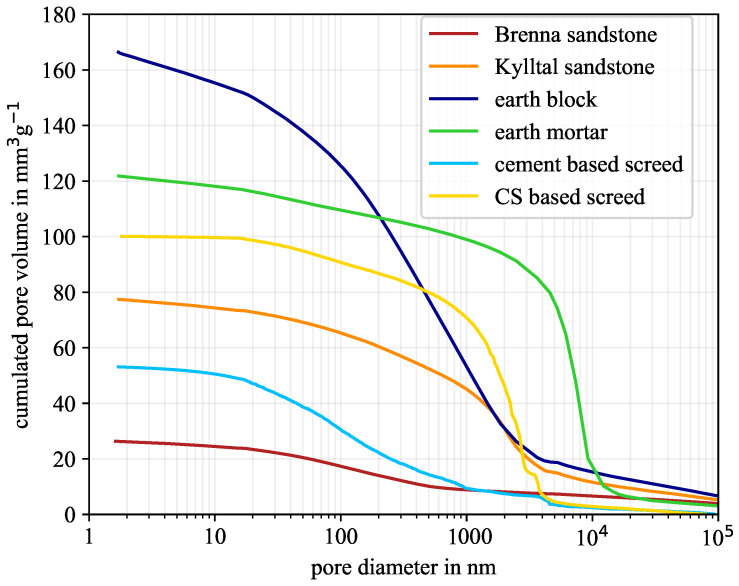
Measured cumulated pore-size distribution based on mercury intrusion porosimetry and gas adsorption. (CS—calcium sulphate).

**Figure 3 molecules-26-07190-f003:**
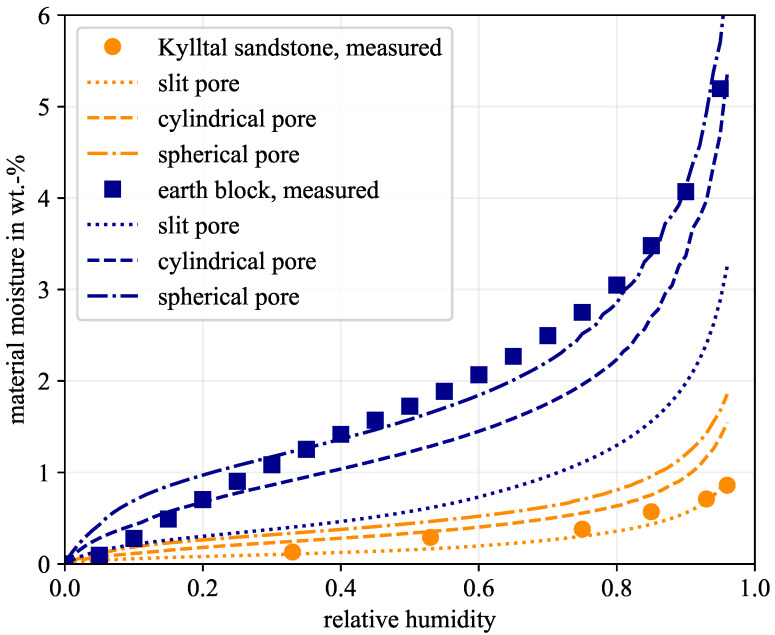
Comparison of the measured and the predicted sorption isotherm for the three investigated pore geometries.

**Figure 4 molecules-26-07190-f004:**
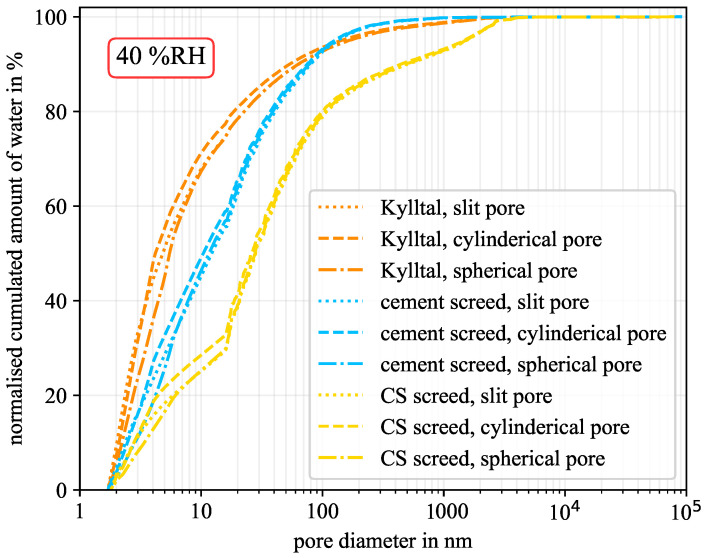
Normalised cumulated amount of water at 40% RH for the three pore geometries. (CS—calcium sulphate).

**Figure 5 molecules-26-07190-f005:**
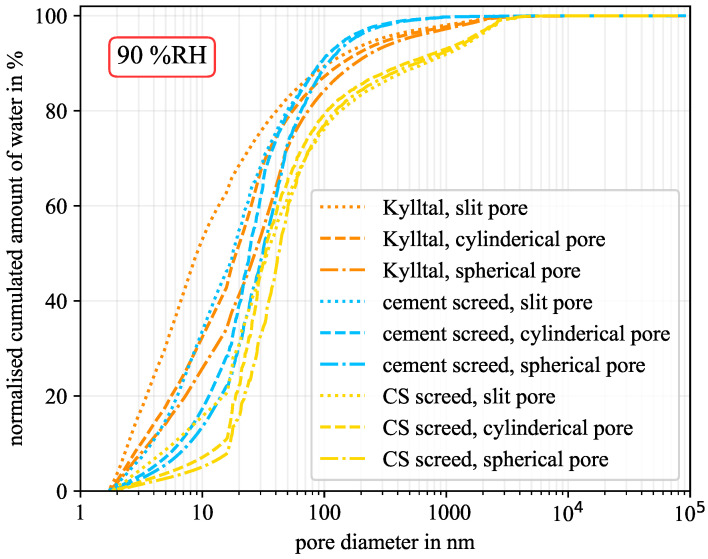
Normalised cumulated amount of water at 90% RH for the three pore geometries. (CS—calcium sulphate).

**Figure 6 molecules-26-07190-f006:**
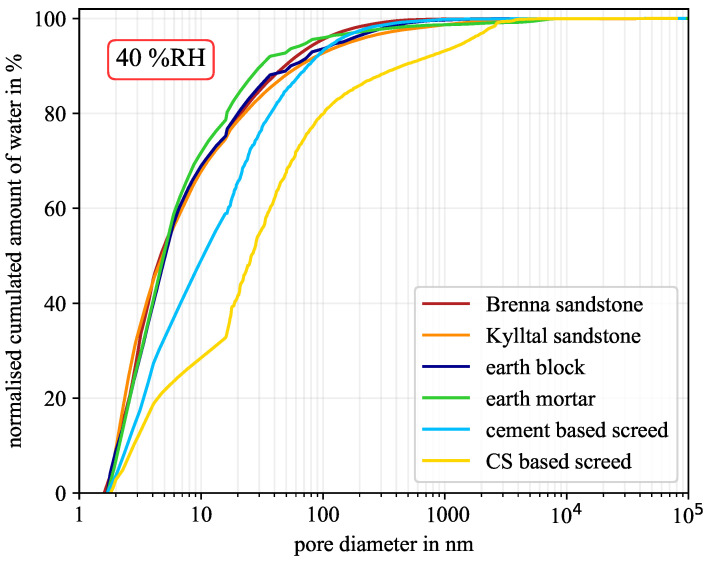
Normalised cumulated amount of water at 40% RH for the six material samples. (CS—calcium sulphate).

**Figure 7 molecules-26-07190-f007:**
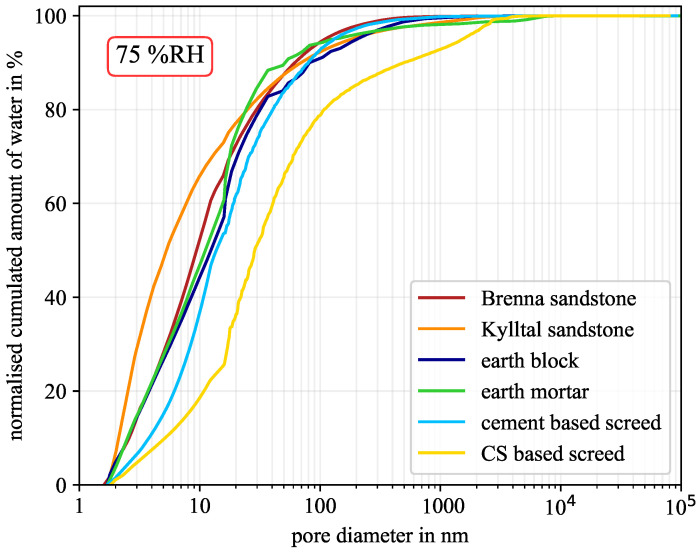
Normalised cumulated amount of water at 75% RH for the six material samples. (CS—calcium sulphate).

**Figure 8 molecules-26-07190-f008:**
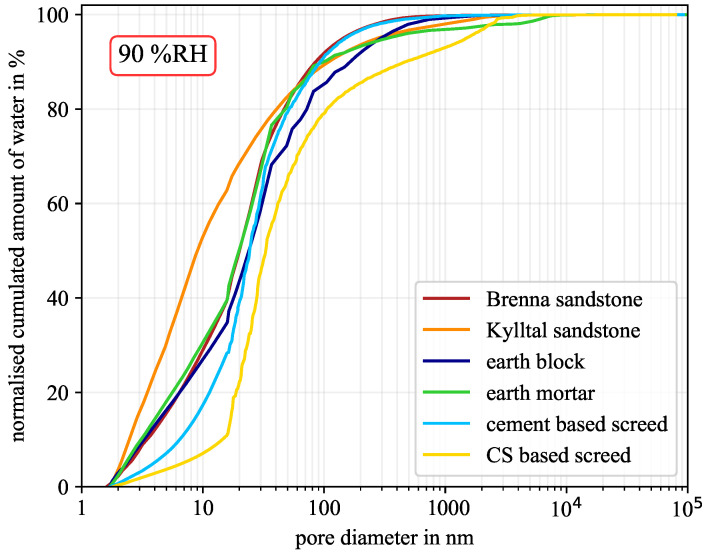
Normalised cumulated amount of water at 90% RH for the six material samples. (CS—calcium sulphate).

**Figure 9 molecules-26-07190-f009:**
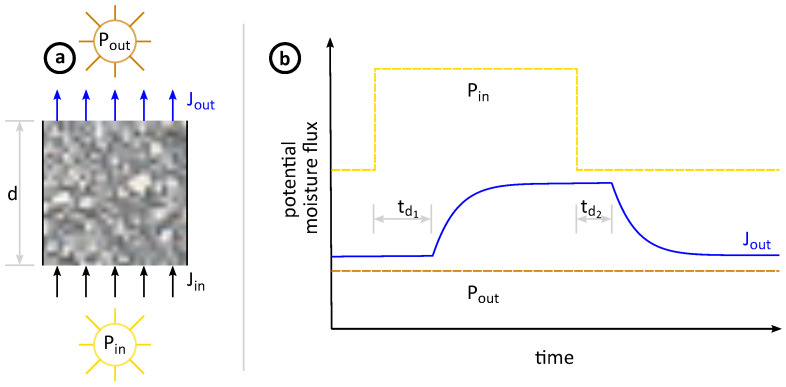
Moisture flux through a thick structure including physisorption.

**Figure 10 molecules-26-07190-f010:**
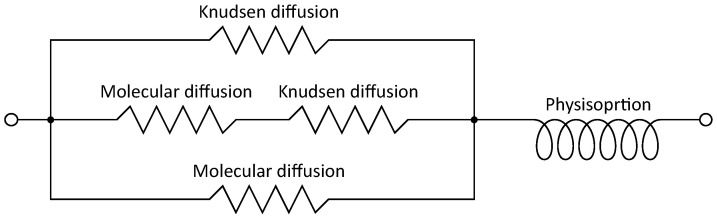
Electrical analogical circuit for the diffusion coefficient of a thick structure including physiportion. The resistor circuit part is adapted from [99].

**Table 1 molecules-26-07190-t001:** Material parameters and properties used for free liquid water and water vapour at 296.15 K and 101,325 Pa.

Parameter	Numeric Value	Unit
*C*	15	
*M*	180,153 $\cdot 10^{-6}$	kg mol${}^{-1}$
*R*	8.314	J mol${}^{-1}$ K${}^{-1}$
γ	72,232 $\cdot 10^{-6}$	Nm${}^{-1}$
$\rho_{air}$	1.19221	kg m${}^{-3}$
$\rho_{l}$	997.5	kg m${}^{-3}$

**Table 2 molecules-26-07190-t002:** Used salts for salt solutions and the regulated relative humidities.

Salt	Relative Humidity in %
magnesium chloride	33
magnesium nitrate	53
sodium chloride	75
potassium chloride	85
ammonium dihydrogen phosphate	93
potassium dihydrogen phosphate	96

**Table 3 molecules-26-07190-t003:** Amount of mesopores within the PSD.

Material	Total Pore Volume	Absolute Amount of Mesopores	Relative Amount of Mesopores
Brenna sandstone	26.3 mm${}^{3}$ g${}^{-1}$	5.9 mm${}^{3}$ g${}^{-1}$	22.4%
Kylltal sandstone	77.4 mm${}^{3}$ g${}^{-1}$	8.2 mm${}^{3}$ g${}^{-1}$	10.6%
earth block	166.4 mm${}^{3}$ g${}^{-1}$	28.2 mm${}^{3}$ g${}^{-1}$	16.9%
earth mortar	121.8 mm${}^{3}$ g${}^{-1}$	9.4 mm${}^{3}$ g${}^{-1}$	7.7%
cement-based screed	53.1 mm${}^{3}$ g${}^{-1}$	14.3 mm${}^{3}$ g${}^{-1}$	26.9%
calcium sulphate-based screed	100.1 mm${}^{3}$ g${}^{-1}$	5.2 mm${}^{3}$ g${}^{-1}$	5.2%

**Table 4 molecules-26-07190-t004:** Chosen pore geometry for the pore network of the six material samples.

Material	Best Fitting Pore Geometry
Brenna sandstone	cylindrical pore
Kylltal sandstone	slit pore
earth block	spherical pore
earth mortar	spherical pore
cement-based screed	cylindrical pore
calcium sulphate-based screed	cylindrical pore

**Table 5 molecules-26-07190-t005:** Moisture located in mesopores compared to the overall moisture.

Material	40% RH	75% RH	90% RH
Brenna sandstone	90.3%	87.5%	81.4%
Kylltal sandstone	88.2%	87.5%	82.7%
earth block	90.6%	86.9%	76.5%
earth mortar	92.9%	89.6%	80.9%
cement-based screed	84.9%	83.8%	79.4%
calcium sulphate-based screed	67.6%	65.5%	65%

## Data Availability

The screed data are partially available on (last access: 21 November 2021): https://pubmed.ncbi.nlm.nih.gov/30310833/ [100]. All other data, including the combined pore-size distribution and the experimental sorption isotherm, are available on a data repository (last access: 21 November 2021): https://doi.org/10.7910/DVN/NILGW2.

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
