# Peer review of "About the Dominance of Mesopores in Physisorption in Amorphous Materials"

_molecules, 2021, doi:10.3390/molecules26237190_

Round 1

Reviewer 1 Report

The authors reported an interesting topic of physisorption of amorphous materials for water adsorption and moisture distribution etc. The manuscript was written well. It is better to include a short discussion for the performance of some important solids, such as MCM-41, MOF, and organic-inorganic hybrid metal-phosphate materials.

Author Response

The authors reported an interesting topic of physisorption of amorphous materials for water adsorption and moisture distribution etc. The manuscript was written well. It is better to include a short discussion for the performance of some important solids, such as MCM-41, MOF, and organic-inorganic hybrid metal-phosphate materials.

Answer: Dear referee, thank you very much for your great feedback. Currently, we are looking for reference materials with regular pore networks. The announced materials have excellent properties. We added in the text the following paragraph:

“Moreover, physisorption is crucial for any combination of adsorbate and adsorbent. Mesoporous molecular sieve materials such as MCM-41 have huge inner surface areas of up to 600 m2/g and show high potential for moisture regulation due to physisorption [48]. The capability of MCM-41 regarding CO2separation from gas mixtures is also influenced by the material moisture [49]. Similar to MCM-41, hydrophilic metal-organic frameworks (MOF) possess large inner surfaces and their physisorption characteristics highly influences the efficiency of adsorption heat pumps and atmospheric water generators [50]. MOFs were tested with other adsorbates such as methanol and ethanol with the purpose to optimise the purification of alcohol-based biofuels [51]. Physisorption of moisture also influences the permeability of CO2and N2in zeolite membranes. The interactions of material moisture and the efficiency of gas separation is not fully understood [52].”

Reviewer 2 Report

The manuscript focuses on investigation of water sorption in porous amorphous materials.  The study highlights the predominant sorption of water vapours in mesopores of amorphous material which is especially important due to the poor knowledge of adsorption mechanisms for amorphous porous materials. Therefore, the research is original and could be interesting for the readership of the Molecules. The manuscript is well-written, the language is concise and understandable.

However, there are some concerns that need to be addressed:

1) In my opinion, it is necessary to provide the elemental composition of the studied compounds. How do you think the chemical composition of the amorphous material is related to the relative amount of mesopores? For example, calcium sulphate screed versus Brenna and Kylltal sandstone

2) Recently, the DFT-based methods for the prediction of sorption isotherms have become widespread due to their greater accuracy than classical sorption models. Please, briefly describe what is the advantage of the models used in the work in comparison with modern DFT methods.

3) In your opinion, what is the significant difference in the composition of Brenna and Kylltal sandstones, which leads to the difference in the preferred geometry of their mesopores (cylindrical and slit pores for Brenna and Kylltal sandstones correspodingly).

Author Response

The manuscript focuses on investigation of water sorption in porous amorphous materials.  The study highlights the predominant sorption of water vapours in mesopores of amorphous material which is especially important due to the poor knowledge of adsorption mechanisms for amorphous porous materials. Therefore, the research is original and could be interesting for the readership of the Molecules. The manuscript is well-written, the language is concise and understandable.

However, there are some concerns that need to be addressed:

1) In my opinion, it is necessary to provide the elemental composition of the studied compounds. How do you think the chemical composition of the amorphous material is related to the relative amount of mesopores? For example, calcium sulphate screed versus Brenna and Kylltal sandstone

Answer: As we don’t have analytic results of the chemical components of the sandstones, we mainly refer to chemical composition descriptions in the literature. However, for sandstones, we think the relative amount of mesopores is mainly influenced by the diagenesis conditions and the sedimentation processes beforehand. In detail, this means the sorting and roundness degree of the grains. Moreover, we think the amount of clay and mica minerals plays also in important role. The Brenna, especially, is a well-sorted sandstone with a low porosity as the grains have many contact areas. A characteristic of the Kylltal sandstone is the high amount of mica, which are layered silica minerals. A scanning electron microscope (SEM) result that is shown in Grimm2018 shows large void spaces that are partially filled with clayey matrix. Nevertheless, for further discussion, we suggest that SEM measurements are needed as well as the expertise of a mineralogist.

In comparison to calcium-sulphate screed, we see differences in the type of porosity that forms. As we mainly have intergranular pores in the sandstones, in hydrating building materials, the forming of C-S-H leads to inter- and intraparticle pores.   

2) Recently, the DFT-based methods for the prediction of sorption isotherms have become widespread due to their greater accuracy than classical sorption models. Please, briefly describe what is the advantage of the models used in the work in comparison with modern DFT methods.

Answer: To be honest, I am not an expert for DFT-based methods. In general, DFT-based methods are developed to characterise materials which incorporate micro and mesopores. Furthermore, several DFTs exist and lead to very different results, caused by the chosen Kernel. Thus, precise foreknowledge is required to model the pore size distribution correctly. If the model is validated by experiments, indeed, it is a very powerful tool because several effects can be quantified separately, such as hysteresis, metastability of confined fluids, pore blocking, and network effects. However, by my best knowledge, I do not know any study which investigate neither water vapour adsorption, nor amorphous (building) materials. Some possible reason I will discuss, but they are speculative, because I am not an expert for DFT:

  • Building materials possess almost no micropores. For example, the C-S-H phase and related gel pores in concrete close almost any pore space below 2 nm. In contrast, macropores are dominating the pore network. Therefore, DFT, mainly for pores below 100 nm, is not able to describe the investigated materials adequately. Macropores cannot be measured vs. micropore do not exist.
  • Regarding the water vapour adsorption: Water vapour is polar. Therefore, the model is very complex because not only the first layer, but the following layers are also influenced by the polarity of the solid (up to the first 5 to 12 layers, depending on the study). This increases the complexity of the Kernel significantly.
  • Inhomogeneity: In contrast to crystalline materials, the surface and its polarity is probably not homogeneous. Several elements and minerals are involved with very different surface charge density. These probably leads to hydration clusters, especially for polar fluids. I do not know, if DFT Kernels can handle this issue. If think, this would be an interesting research question.
  • Pore network: One has to know a priori, which Kernel can be used for a distinct material. But materials such as concrete, sandstone, clay, stones, bricks, etc. will never be identically. Although the basic material might be identical, slight differences in temperature, moisture, or pressure will results in a modified pore network. Thus, the Kernel of the DFT must be adapted for each sample. Of course, this is manageable, but not our intension. Our aim was to develop and validate a robust and general model to describe material moisture in amorphous materials.

3) In your opinion, what is the significant difference in the composition of Brenna and Kylltal sandstones, which leads to the difference in the preferred geometry of their mesopores (cylindrical and slit pores for Brenna and Kylltal sandstones correspondingly).

Answer: As described in the answer to questions 1, both sandstones have a clayey matrix and mainly consist of quartz and rock fragments. But a difference is the roundness degree of the grains, which are less rounded in the Brenna sandstone. Another difference is the large amount of mica minerals in the Kylltal sandstone, which may lead to a more flat pore geometry

We added the following text passage into section 4.1:

The amount of mesopores differs significantly, e.g. by 17.2~\% between the calcium sulphate-based screed and the Brenna sandstone. Consequently, the question arises what causes these great differences. There is not a unique influencing factor. Instead, the authors speculate that the complex combination of chemical composition and forming condition determines the pore network (sedimentation and diagenesis conditions of the sandstones, hydration process of the screed and earth material, etc.). In the case of screed, we think that the formation of calcium silicate hydrates may lead to a totally different pore geometry and network compared to sandstones. However, detailed analysis of the chemical composition is out of the scope of this study.

We added the following text passage into section 4.2:

The fitting of the pore geometry is just an ordinary correlation without the incorporation of the material physics. This is mainly caused by the fact that the "true" pore system is unknown. However, within the group of the sandstones, the bet fitting pore geometry varies. One speculative explanation is that the Kylltal sandstone contains mica minerals and a large amount of clayey binders. In scanning electron microscope results, these minerals look more rodlike and therefor might cause a more slit-shaped pore geometry \cite{grimm2018bildatlas}. However, a detailed discussion between assumed pore geometry and tomographic imaging is out of the scope of this study.

Reviewer 3 Report

The paper presents research on the determination of mesopores and PSDs from physisorption in amorphous materials. The presentation of methods and scientific results in the current form is unsatisfactory for publication in the Molecules journal. The minor and significant drawbacks to be addressed can be specified as follows:
1.    Fig. 1. Is there a correlation between maximum water layer thickness and %RH for a given type of pore geometry? For which minimum pore widths the curves were plotted? Also for micropores?
2.    Page 6, 3.4. Mercury intrusion porosimetry and gas adsorption. What apparatus? For which pore widths can the BJH method be used? For micropores? Hmmmm .... What shape of pores was adopted in the BJH method? Slits? Cylinders? Of course, the choice is crucial!!!
3.    Fig. 3. It is a pity that water isotherms were not shown for the remaining materials. Have the authors considered a mixed type of pores? For example, for earth block: narrow pores – cylinders and for wider pores  - spheres.
4.    Using the dual notation "CS based screed" and "calcium sulphate-based screed" is sometimes confusing.
5.    How does "Normalized cumulated amount of water" relate to "cumulated pore-size distribution based on mercury intrusion porosimetry and gas adsorption"? The same pore structure is detected by these methods.
6.    The short biography of Philipp Wiehle is missing. ;)

Author Response

The paper presents research on the determination of mesopores and PSDs from physisorption in amorphous materials. The presentation of methods and scientific results in the current form is unsatisfactory for publication in the Molecules journal. The minor and significant drawbacks to be addressed can be specified as follows:

  1.    Fig. 1. Is there a correlation between maximum water layer thickness and %RH for a given type of pore geometry? For which minimum pore widths the curves were plotted? Also for micropores?

Answer: This is exactly we were trying to show with this plot: There is a maximum water layer thickness for the different pore geometries in dependence to the relative humidity. Due to the curvature of the fluid’s surface, the equilibrium water vapour pressure is increased. The higher the curvature, the more water can be retained. The effect is quantified by the Kelvin equation. Consequently, spherical pores hold more material moisture than cylindrical pores for a certain humidity level.

The minimum pore width for this plot was 1.7 nm in diameter. Below this diameter, we do not have data of the pore size distribution. However, to clarify the applicability of the theory, we added the following sentence in section 2.2.:

“The Kelvin equation is considered to be applicable down to a capillary diameter of 1 nm [56].”

Following the given reference, micropores down to 1 nm can be considered as well by the theory.

  1.    Page 6, 3.4. Mercury intrusion porosimetry and gas adsorption. What apparatus? For which pore widths can the BJH method be used? For micropores? Hmmmm .... What shape of pores was adopted in the BJH method? Slits? Cylinders? Of course, the choice is crucial!!!

Answer:

Apparatus: We added the names of the used apparatuses:

‘by the measurement device 'MicroActive AutoPore V 9600' … ‘The measurements were performed by the device 'ASAP 2010 V5.03'’’

Pore width: Following the ISO 15901 standard, mesopores and macropores are analysed for diameter approximately between 2 nm to 100 nm.

Micropores: Following the ISO 15901 standard, gas adsorption is able to characterise micropores in the range of 0.4 nm to 2 nm as well. This analysis was not done in our measurement device. The determined minimum pore diameter was 1.7 nm, as shown in figure 2.

Regarding the chosen pore geometry, we added the geometry in the end of the sentence in the text:

‘The conversion from pressure to a certain pore diameter followed the Barrett-Joyner-Halenda (BJH) theory by assuming cylindrical pores’

  1.    Fig. 3. It is a pity that water isotherms were not shown for the remaining materials. Have the authors considered a mixed type of pores? For example, for earth block: narrow pores – cylinders and for wider pores  - spheres.

Answer: Indeed, the sorption isotherms are not shown, but all data (including the sorption isotherms) are available in the data repository. Regarding the mixed type model: This would rise the question, which is the “correct” threshold to separate between narrow and wider pores. This seems to be a little bit arbitrary. Of course, one might fit this with a kind of least square regression. But in my eyes, doing so, we are leaving the analytical approach and move to empirical curve fitting of sorption isotherms. Thus, this reduces the generality of the model drastically.

A second thought. As shown in figure 1, the curvature effect (already discussed in answer 1) is significantly reduced/ negligible for pore diameter larger than 100 nm. Therefore, for wider pores, it is almost irrelevant, which pore model is chosen. The important part are the mesopores.

  1.    Using the dual notation "CS based screed" and "calcium sulphate-based screed" is sometimes confusing.

Answer: I do not like is either. We used CS just in the legends, because elsewise, the legend is blocking graphs. However, to make it more reader friendly, we added in the legend in the relevant plots.

“(CS - calcium sulphate)”

We hope, this avoids confusion.

  1.    How does "Normalized cumulated amount of water" relate to "cumulated pore-size distribution based on mercury intrusion porosimetry and gas adsorption"? The same pore structure is detected by these methods.

Answer: Of course, the underlying pore system is every time the same, hence, the used pore size distribution is constant. For a distinct pore radius (e.g. 10 nm) and humidity (e.g. 75 %RH), due to the curvature of the water film, the water layer thickness depends on the pore geometry (please see figure 1). If we would cumulate these water amounts, we would end up with different levels of material moisture (which are the sorption isotherms showing in figure 3). This would make a comparison quite difficult. Therefore, we normalize these curves with the total amount of water for each considered humidity levels. We already tried to express this with the sentence in section 4.3:

‘For a better comparison between samples, the cumulated amount of water was normalised by the total amount of water for each pore geometry of each sample.’

We do see the current explanation is probably too short to follow. Therefore, we extended this, as you can see in the following.

‘Thus, for a certain humidity level, the material moisture differs between the investigated pore geometries. However, starting the cumulation at the smallest diameter of 1.8~nm, one can see how much water is held in the pores. Thereby, spherical pores would show higher amounts of water due to the thicker water layer caused by the great curvature. For a better comparison between samples, the cumulated amount of water was normalised by the total amount of water for each pore geometry of each sample at the considered humidity level.’

  1.    The short biography of Philipp Wiehle is missing. ;)

Answer: Thanks for this, we were already aware of this. The biography is added.

Round 2

Reviewer 1 Report

I love this revised manuscript. This also reminds me of several interesting examples of germanates, metal phosphates (or metal phosphates) with large pores (up to 36-, 28-, and 72- membered rings, respectively). It is going to be perfect if these compounds can be included in this manuscript. 

Author Response

Thank you very much for further ideas of reference materials. We added to the text:

“Also, mesoporous germanate is used for CO2separation and, depending on the configuration, possesses a bimodal pore size distribution [53]. Due to distinct changes in electrical impedance due to physisorption of water vapour, mesoporous materials such as metal phosphates might be used directly as humidity sensor [54].”

Reviewer 3 Report

Congratulations on a great job. The author has made a substantial improvement for this article. The manuscript can be accepted for publishment in the present form.

Author Response

Thank you very much for your valuable and detailed input. 
